# The Impact of Responsible Food Packaging Perceptions on Naturalness and Healthiness Inferences, and Consumer Buying Intentions

**DOI:** 10.3390/foods10102366

**Published:** 2021-10-05

**Authors:** Alain d’Astous, JoAnne Labrecque

**Affiliations:** HEC Montréal, Montréal, QC H3T 2A7, Canada; joanne.labrecque@hec.ca

**Keywords:** food packaging, food naturalness, food healthiness, food buying intention

## Abstract

The research presented in this article examines the relationship between consumer perceptions that a food package is responsible (sustainable) and consumers’ intention to purchase the product that it contains. On the basis of the relevant literature, a conceptual model is proposed where this relationship is hypothesized to be mediated by two variables: the product’s perceived naturalness and healthiness. A first study was conducted with the objective of developing a scale with good psychometric properties to measure the perceived naturalness of a food product. The objective of the second study was to test the validity of the two-mediator conceptual model. The results show that the extent to which a food product package is seen as responsible (i.e., recyclable, reusable, compostable) has a positive and statistically significant impact on consumers’ intention to buy it, and that it is through the sequential mediation of the product’s perceived naturalness and healthiness that this relationship unfolds.

## 1. Introduction

Over the last thirty years, consumers have gradually increased their consumption of processed foods and ready meals [1,2,3], in a quest for efficiency and convenience. This greater demand has resulted in a significant increase in processing methods in the food chain and, consequently, accentuated consumer concerns about food technologies, food production and food processing methods. Consumers question the safety and the content of the foods they consume on a regular basis and the harm the food industry causes to the environment. According to Eaters Digest (2017) [4], less than forty percent of the participants in their worldwide survey trust the food industry to provide them with healthful food, and seven out of ten participants reported that they ‘worry that our food supply is becoming increasingly contaminated/unsafe’ (p. 11). In addition to convenience, more consumers are considering health and sustainability in their food choice, as demonstrated by the greater demand for organic [5,6], local and natural food products [1,7,8,9]. This growing interest in ‘natural’ products has led many consumer researchers to study the topic of consumers’ perception of food naturalness and its impact on behavior.

Research has explored what food naturalness means for consumers. In general, consumers consider natural entities to be healthier, more appealing and better for the environment [10]. For consumers, defining the characteristics of what they consider to be a “natural” food is however ambiguous. They often define food naturalness by what it is not. Thus, in their evaluation of food naturalness, consumers refer to unnatural aspects such as artificial ingredients, additives, in addition to referring also to natural characteristics such as having minimal or no human intervention [11,12]. These factors influence their attitudes toward products. Rozin and colleagues [10,11,12,13,14] have shown that perceptions of food naturalness are minimally affected by mixing like entities and are more compromised by chemical transformations than by physical transformations. Moreover, they affirm that it is the extent of food processing that has the greatest negative effect on consumers’ perception of food naturalness.

In their systematic review of the research on the significance of food naturalness among consumers, Roman et al. (2017) [15] noted significant differences in the definitions and measurements of food naturalness across studies. Based on their review, they proposed a list of items aimed at measuring perceived food naturalness that attempts to take into account the multifaceted structure of the construct. Their three-category series of items encompasses: (1) how the food is grown (e.g., organically, locally); (2) how the food is produced (i.e., technology used and the presence of desirable and undesirable components such as artificial ingredients, preservatives, additives, artificial colors and flavors, chemical hormones and pesticides, GMOs); and (3) what the properties of the final products are (i.e., healthiness, friendliness, taste, freshness). The items are presumed to include the many criteria that can be considered when evaluating food naturalness from consumers’ standpoint. It is worth noting that the items proposed by Roman et al. (2017) reflect not only the natural properties of the food product itself (i.e., the core concept), but also the determinants of food naturalness (e.g., methods of growing, production process).

Food product evaluations are usually based on inferences that consumers draw from their understanding and interpretation of informational cues on packages [16] as well as from their attractiveness [17,18,19]. Numerous studies conducted by marketing researchers have established the influence of packaging information related to various product characteristics on consumer product evaluation and purchase behavior, notably information that relates to nutrition and health (e.g., labels, claims) [19,20,21,22,23], ingredients [16,24,25,26,27,28], processing methods [29,30,31], organic content and sustainability [32,33,34], and fair trade [35].

Research has also examined the impact of various attributes of food packaging on the perceived quality, healthiness and naturalness of its contents. In their extensive review of studies on the effect of food packaging visual cues on consumer behavior, Vermeir and Roose (2020) [36] considered the influence of visual cues on behavioral outcomes and identified research gaps in the food context. Their analysis is based on the visual cue typologies of Adaval et al. (2018) [37] and Sample et al. (2020) [38] and, consequently, their discussion is limited to the impact of prominent visual cues. They reported that the color of the packaging (i.e., hue, lightness and saturation), its shape (i.e., dimensionality, demarcation and completeness) and texture (i.e., packaging material) all contribute significantly to perceptions of food quality and healthiness, depending on the product type and the context. Aesthetic cues, especially the complexity of the packaging design, also influence quality perceptions. In their discussion, Vermeir and Roose (2020) [36] highlighted the lack of research on the effects of packaging visual cues on sustainability perceptions and behavior.

Results from recent studies indicate that types of nutrition claims such as ‘low fat’ or ‘free of GMOs’ [26,39], ‘natural’ claims [40], production methods [41], and package appearance [42,43,44] influence attribute-based inferences about healthiness. Frizzo et al. (2020) [41] concluded that naturalness plays a mediation role between the production method and consumers’ intentions. Berry et al. (2017) [40] found similar results for the ‘natural’ claim. Finally, Steenis et al. (2017) [45] analyzed the impact of consumer evaluation of packaging sustainability on product associated benefits (i.e., convenient, healthy, natural, quality). They found a significant relation between sustainability perceptions of packaging and all benefits. However, consumer perceptions of green packaging are not always clear and consumers have different motives for choosing green-packaged products [46]. Otto et al. (2021) [47] also pointed out that consumer perception of packaging materials differs from the objective assessment of packaging environmental sustainability.

Given growing environmental considerations, a particularly interesting question concerns the link between consumers’ perceptions of packaging sustainability and naturalness and health inferences, as well as their impact on food purchase intention. The objective of this research is therefore to examine this question empirically.

## 2. Conceptual Development

The conceptual model that has guided this research is displayed in Figure 1. This model follows from the literature review presented in the preceding section. Its objective is to present a theoretical explanation of the presumed positive relationship between the perceived responsible (sustainable) character of a food package and the intention to buy the food product that it contains.

As can be seen in Figure 1, this theoretical explanation is founded on the sequential mediation effects of two variables: the food’s perceived naturalness and healthiness. More precisely, the extent to which a food package is perceived as responsible (i.e., recyclable, reusable, compostable) is hypothesized to have a positive impact on the perceived naturalness of the food product. This theoretical relationship is based on the studies by Steenis et al. (2017) [45], Marckhgott and Kamleitner (2019) [43], and Vermeir and Roose (2020) [36], among others. In turn, the perceived naturalness of the food product is presumed to have a positive impact on its perceived healthiness. This theoretical relationship follows from research conducted by Rozin et al. (2012) [14] and Binninger (2017) [42]. Finally, the model proposes that the greater the perceived healthiness of the food product, the stronger consumers’ intention to buy it. These theoretical relationships lead us to posit the following general research hypothesis:

**Hypothesis** **1** **(H1).**
*The degree to which a package that contains a food product is perceived as responsible (recyclable, reusable, compostable) has a positive effect on consumers’ intention to buy the product. The relationship is explained by the sequential mediation effects of the food product’s perceived naturalness and healthiness, in that order.*


## 3. Study 1

The objective of Study 1 was to develop a scale aimed at assessing the perceived naturalness of food products.

### Method and Results

The study consisted in three phases. In phase 1, a list of synonyms of the word “natural” was produced with the help of a dictionary and different online search engines. This resulted in about 40 different words that seemed to cover adequately the meaning of the naturalness concept as it would apply to food. Thus, synonyms such as spontaneous, normal, frank and the like that were proposed in the dictionary or generated by any of the search engines were a priori dismissed since they clearly referred to irrelevant meanings in the context of this research.

The objective of the second phase was to carry out a first epuration of this preliminary list of synonyms. This phase entailed the participation of five adult female consumers. Although these persons formed a convenience sample, they all displayed true interest in buying and eating healthy foods. They gathered in the home of one of the participants with their own microcomputer. The interviewer e-mailed them the list of synonyms as an Excel file. Their task, which was performed individually, was to think of some natural food products (e.g., cereals products, dairy products, prepared foods) and to indicate which of the terms in the list would *not* qualify as synonyms of the word natural in this particular context. The participants’ responses were grouped in a single file and, as a decision rule, the terms which obtained four votes or more were deleted from the original list. This resulted in a reduced list of 23 synonyms.

The objective of the third phase was to further reduce the number of synonyms. This was accomplished by means of an online survey with a sample of adult consumers. This sample is composed of persons recruited from the researchers’ social networks. From a total of about 180 email invitations to participate, 101 usable questionnaires could be collected. The sample comprises a majority of women (57.4%) and the age of the participants varies from 19 to 82 years (mean = 34.7). To the question “How important is it for you to consume natural products?”, 66.3% said it was very important and 20% that it was extremely important. These data tend to confirm that the respondents are very involved in the consumption of natural products.

The survey participants were asked to indicate on a 5-point scale the probability (from unlikely to very likely) of using each term in the list of synonyms when thinking of natural foods. Using a mean of 3.5 as a cut-off point, a first reduced list of 11 terms was obtained: natural (mean = 4.71), healthy (4.57), fresh (4.42), local (4.27), biological (4.17), ecological (3.85), domestic (3.71), essential (3.62), authentic (3.57), tasty (3.57), and simple (3.52). The results of a factor analysis conducted with these 11 terms confirmed that they formed a single factor [48] Cronbach’s alpha coefficient was equal to 0.70. To further reduce the list and enhance reliability, each item whose elimination contributed to increase that value was excluded until no more increase was observed. This led to the elimination of five terms that correspond exactly to those having the lowest mean subjective probability of being used in the 11-term list. The final scale is therefore composed of the following items: natural, healthy, fresh, local, biological and ecological, and its Cronbach’s alpha value was equal to 0.72 in this sample.

## 4. Study 2

### 4.1. Overview

The objective of Study 2 was to test the adequateness of the conceptual model (Figure 1). To this end, a sample of 120 adult consumers participated in a “food product evaluation study” where they were asked to rate in sequence five food products, out of a total of 20 different product stimuli, as regards their buying intention, the extent to which they perceived that the products were consistent with healthy eating, their perceived naturalness, and the responsible nature of their packaging, in that order.

### 4.2. Method

#### 4.2.1. Data Collection Site

In order to maximize the realism of the study, real food product stimuli were used. They were made available in a room located in the researchers’ university. Products that required to be kept cool (e.g., dairy products) were placed in a refrigerated display whereas the others were put on tables. As a whole, the room was meant to look like a small food store (see Appendix A).

#### 4.2.2. Product Stimuli

The size of the universe of food products is enormous and any attempt at coming up with a representative sample of cases is hopeless. As a more modest approach, it was deemed important to assemble a set of products exhibiting good variance as regards the product categories, the types of packaging, and the brands. To reach this goal, four product categories were identified: dairy products (six different products covering milk and milk substitute, yogurt, and cheese), meats (four different products covering ham, and chicken breasts), cereals products (four different products covering bread, and tortillas), and fruits and vegetables (six different products covering fresh vegetables, canned vegetables, and fruit salads). More detailed information on the product stimuli appears in Appendix B.

#### 4.2.3. Sampling

Written invitations to participate in a university study were distributed in the neighborhoods surrounding the researchers’ university as well as among university employees. The objective was to get a diversified, although not probabilistic, sample of people that would form a typical group of food products consumers. In order to encourage enrollment, the invitation mentioned that they would receive 20 $ in cash, some coupons, and a 20 $ gift card redeemable in a nearby bakery. The would-be participants had to call a given phone number to schedule a specific time for their visit to the study room. The sampling process ended when a sample size of 120 participants was attained.

#### 4.2.4. Data Collection

The data were collected one person at a time. Upon entering the local, the person was informed of the task, namely that she or he would have to answer a series of questions regarding five different food products. The five products and their order of presentation were determined a priori by the researchers such that the set comprised at least one product from each product category. The product stimuli were selected in a random fashion with the objective of having each product rated by 30 participants (30 participants × 20 = 600 evaluations in total = 120 participants × 5 evaluations). To avoid any order effect associated with the product categories, the sequence of food products presented to the participant was varied systematically.

#### 4.2.5. Measures

The participants were asked to take in hand and examine the first of the five food products that were assigned to them. When their examination was completed, they filed in a one-page questionnaire containing measures of the theoretical concepts. This process was repeated with the four remaining products. Upon completing this evaluation task, the participants filled in a questionnaire with several descriptive measures (see the sample description below).

The one-page product evaluation questionnaire contained the measures of the theoretical concepts. Intention to buy was assessed with a single-item scale: “In the event that you would need to buy this type of product, what is the probability that you would choose this product rather than a different one?” The participant had to circle a percentage from 0% to 100% presented in 10% increments (i.e., 0% 10% … 100%). The perception that the product was consistent with healthy eating was assessed with two items: “Consuming this product is consistent with healthy food taking” and “In the perspective of healthy eating, the consumption of this product seems appropriate to me”. Each item was rated on a 7-point bipolar numerical scale with endpoints totally disagree/totally agree. Perceived naturalness was measured with the 6-item scale developed in Study 1: biological, fresh, local, healthy, natural, and ecological. Each term was rated on a 7-point bipolar numerical scale with endpoints not at all/completely. The responsible nature of the packaging was measured with three items: recyclable, reusable, and compostable (7-point not at all/completely numerical scale).

### 4.3. Results

#### 4.3.1. Sample Description

The sample is composed of a majority of female participants (65.8%), which is in line with survey data showing that women form the primary group of grocery shoppers [6]. To the question “Are your responsible for doing the grocery shopping in your household?”, the mean is 5.26 on a seven-point numerical scale with endpoints never/always, a result which confirms that the sample as a whole is relevant with respect to the objectives of this study. The age of the participants varies from 18 to 67 years with a mean of 44.4. They are well educated, as 62.5% reported that they had some university education. The annual household income before taxes shows great variance since nearly 30% of the participants reported 60,000 $ or less and nearly 30% 100,000 $ or more. A majority of the participants (66.7%) are full-time workers.

#### 4.3.2. Creation of the Variables and Psychometric Checks

As mentioned previously, each participant provided responses for five different products out of a total of 20 product stimuli. The mean ratings of all study variables as regards these five products was computed for each participant. Thus, the final measure of the intention to buy variable reflects the mean intention computed across each respondent’s set of five products. Similarly, the items assessing the product’s perception of healthiness correspond to this perception computed across the respondent’s set of five products. And so on for all theoretical variables.

The correlation between the two items measuring the product’s perceived healthiness is equal to 0.74 and is statistically significant (*p* < 0.01). Therefore, the mean of the scores was taken as an indicator of the product’s health perception. A factor analysis of the items aimed at measuring the product’s perceived naturalness produced a single factor explaining a fair proportion of the total variance (50.4%). The reliability of the final scale is very good (α = 0.77). The mean of the final items served as an indicator of the naturalness concept. The responsible nature of the product’s package was assessed by computing the mean of the three final items: recyclable, reusable, and compostable. Because this corresponds to a formative measure, its reliability was not assessed [49]. The correlations between the study variables are displayed in Table 1.

#### 4.3.3. Mediation Analyses

The conceptual framework of this research represents a two-mediator model in which the effect of the independent variable (responsible packaging) on the dependent variable (buying intention) is presumed to be mediated in sequence by the product’s perceived naturalness and perceived healthiness. This model was tested using template 6 of the PROCESS macro [50]. The estimation procedure entails the regression of naturalness on responsible packaging (Model 1), the regression of perceived healthiness on both naturalness and responsible packaging (Model 2) and the regression of buying intention on the three other variables (Model 3). The results of these regressions are displayed in Table 2.

The regression results that are shown in Table 2 confirm that the responsible nature of the packaging has a positive and statistically significant effect on the product’s perceived naturalness (Model 1) which, in turn, has a positive and significant impact on the product’s perceived healthiness (Model 2) which, in turn, is positively and significantly related to buying intention. As expected, the effects of the independent variable and the mediators are limited to the variable that they directly predict. 

The indirect effects of the responsible nature of the packaging on buying intention were estimated with PROCESS using 5000 bootstrap samples. The only statistically significant indirect effect is that which concern the theoretical sequence presented in Figure 1, that is, responsible packaging → perceived naturalness → perceived healthiness → buying intention. The estimated indirect effect is equal to 2.42 with a 95% confidence interval: [1.41, 4.04]. All other paths from responsible packaging to buying intention are not statistically significant. These results therefore strongly support the theoretical two-mediator model underlying this study.

#### 4.3.4. Additional Analyses

Additional statistical analyses were conducted in order to eliminate some alternative explanations for the results. First, one could argue that firms selling natural food products are more inclined to use responsible packaging. If this is the case, then the relationship between the responsible nature of the product’s package and its perceived naturalness observed in this study would simply reflect this structural commercial strategy, and not the real impact of responsible packaging on perceived naturalness.

In order to verify the plausibility of this explanation, each of the food product stimuli used in this study was examined with the objective of identifying explicit mentions of naturalness on the package (e.g., natural source of vitamin D). Presumably, the package of food products that are natural should logically mention this important characteristic. The package of four of the 20 food products was found to exhibit one such mention while one package had two mentions. The mean of the index of responsible packaging is however higher in the group of products with no mention related to naturalness (mean = 3.49) than in the group with at least one mention (mean = 3.13), although this difference is not statistically significant (*p* = 0.46). This result contributes to eliminate the alternative explanation and supports the theoretical proposition that the extent to which a product’s package is perceived as being responsible contributes positively to the perception that the product is natural (even if it is not).

The positive relationship observed between the responsible nature of the package and the product’s perceived naturalness could perhaps be attributed to the presence of the item “ecological” in the scale assessing this latter concept. In addition, the presence of the item “healthy” in this scale could perhaps explain the positive relationship between this measure and the perceived healthiness of the product. In order to verify the likelihood of these alternative explanations, the mediation analyses presented above were conducted using a measure of perceived naturalness excluding these two items (i.e., taking the mean of biological, fresh, local, and natural). The results of the mediation analyses remained essentially the same, with a positive and statistically significant indirect effect of responsible packaging on buying intention through the sequential mediation of perceived naturalness and perceived healthiness (estimated effect = 2.09 with a 95% confidence interval: [1.15, 3.68]). These results therefore do not support the alternative explanations mentioned above.

## 5. Discussion

The results of this research contribute to enriching the literature on food consumption in two principal ways. The first contribution of this research lies in the conceptual model that has been put forward to explain how consumers’ intention to buy a food product is positively affected by the perceived responsible nature of the package in which the product is presented. Although previous research has shown that extrinsic attributes such as the packaging of a food product may have a significant impact on consumer perceptions related to various intrinsic attributes of the product (e.g., freshness, taste) [36,39,45] and ultimately on purchase behavior [29,42], the conceptual model that has guided this research offers a comprehensive explanation of the psychological mechanisms that consumers deploy when they contemplate the possibility of buying the food product, namely through the presumed package’s impact on the formation of perceptions regarding both the naturalness and healthiness of the product.

The second contribution of this research is methodological. Whereas research that has examined the impact of various attributes of food products on consumers’ perceptions and behavior—whether these attributes are intrinsic (e.g., ingredients, freshness, fat content) or extrinsic (e.g., brand, price, package)—has generally used abstract stimuli in the form of lists of attributes or scenarios [11,12,13,39], the present research has attempted to maximize the ecological validity of the results by using a data collection method (i.e., consumers examining real products that they hold in their hands) that maximally mimics the context in which consumers form their perceptions of food products and their purchase intention.

Nonetheless, this study has limitations that should be kept in mind when interpreting the results. These limitations also represent interesting opportunities for future research. First, although the data collection method has contributed to increasing the realism of the product evaluation context, it should be noted that the participants of this research were asked to look attentively at the product stimuli before answering questions related to the theoretical concepts. It is not clear that when they are in their habitual commercial environments (e.g., grocery stores), consumers adopt such an analytical stance. For instance, when the food products that consumers consider buying have been purchased numerous times in the past, it is unlikely that they will spend much time examining the package and the information displayed on it. Future research should explore the impact of food packaging on consumers’ perceptions and purchase intentions using methods of data collection that are less restrictive with respect to how research participants interact with food product stimuli.

Second, the conceptual model of this research implicitly assumes that consumers spontaneously form perceptions of the responsible character of food product packaging. Indeed, being able to answer questions concerning the likelihood that a package is recyclable, reusable, and compostable does not necessarily imply that such perceptions were formed during the process of examining the product stimuli. Therefore, before concluding that the responsible nature of a food product package has a significant impact on buying intentions, research is needed to explore the spontaneous inferences that consumers make about food products during the buying process. In addition, research should study the effectiveness of various strategies aimed at making the responsible nature of food product packages (e.g., mention that the package is recyclable, mention that the package is made of XX% of recycled materials) salient on consumer perceptions and buying intentions.

The third limitation of this research also concerns the data collection method. While it is definitively relevant to study how consumers react to real food products in environments that resemble the physical places where such products are usually purchased, it must be acknowledged that more and more consumers buy food products online [51]. It would therefore be important to replicate this research using product stimuli that are presented in an electronic format, and to examine the possible differences that emerge when this type of environment is contrasted with that used in this research.

Lastly, it is important to realize that a food product’s offer comprises several features (e.g., overall attractiveness, informational cues, package design) which have an impact on consumers’ evaluation of the product’s contents and, ultimately, on their buying intentions. It is difficult to isolate the effect of a particular aspect of the offer without taking into account the presence of these other features. The extent to which the physical characteristics of the package, the information that it displays, and the perceived sustainable properties of the package are consistent is likely to be a significant factor in the formation of consumer evaluations and intentions to buy. Research is needed to understand how consumers process the totality of the information that is available about food products and how they deal with features that may not be entirely consistent.

## 6. Conclusions

Research on the impact of the perceived naturalness of food products has made it clear that consumers give great importance to this attribute and that it is likely to orient their food choices. The results of this research concur with this general conclusion since the participants’ perceptions as regards the naturalness and healthiness of food product stimuli from several product categories led to increased buying intentions. However, it is not possible for all food products to claim that they are natural, even though many of these products may provide very positive health benefits. In such situations, the results of this research suggest that impressions of naturalness among consumers may nevertheless be created through the use of responsible packaging. When this is possible, communicating adequately the presence of this characteristic to potential consumers may represent an effective commercial strategy for food producers.

This research has focused on one element of a food product’s offer, that is, its package. However, other aspects of a food product’s offer which are distinct from the product contents itself may also contribute to creating impressions of naturalness and healthiness among consumers. For instance, marketing communication about a food product (e.g., advertising, point-of-purchase displays, brand name) may use images and words that lead to inferences of naturalness. Also, ads for a food product may be positioned in diffusion outlets (e.g., Web sites, magazines) that are associated with nature (e.g., a trekking blog). Promotion techniques aimed at increasing the sales of a food product may incorporate elements that relate to nature and health (e.g., a promotional contest to win a guided tour for nature lovers). Research is needed to test the effectiveness of such communication strategies in enhancing naturalness and healthiness food perceptions among consumers.

## Figures and Tables

**Figure 1 foods-10-02366-f001:**
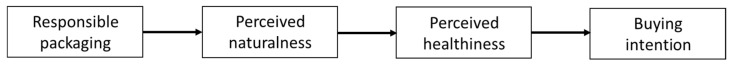
The conceptual model.

**Table 1 foods-10-02366-t001:** Correlations among the study variables.

Variable	M	SD	1	2	3
1 Buying intention	57.67	15.76			
2 Perceived healthiness	5.11	5.11	0.60 *		
3 Perceived naturalness	4.59	4.59	0.51 **	0.69 **	
4 Responsible packaging	3.40	3.40	0.31 **	0.38 **	0.48 **

Note: (*N* = 120); * *p* < 0.05; ** *p* < 0.01; M = mean, SD = standard deviation.

**Table 2 foods-10-02366-t002:** Regression results for mediation analysis.

Model	Dependent Variable	Independent Variables	Regression Coefficient	R^2^
1	Perceived naturalness	Responsible packaging	0.50 ***	0.23 ***
2	Perceived healthiness	Perceived naturalness	0.76 ***	0.48 ***
		Responsible packaging	0.07	
3	Buying intention	Perceived healthiness	6.46 ***	0.38 ***
		Perceived naturalness	2.64	
		Responsible packaging	0.88	

Note: (*N* = 120); *** *p* < 0.001.

## Data Availability

The data presented in this research are available on request from the corresponding author. The data are not publicly available, due to privacy and ethical reasons.

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
