# Peer review of "The Impact of Responsible Food Packaging Perceptions on Naturalness and Healthiness Inferences, and Consumer Buying Intentions"

_foods, 2021, doi:10.3390/foods10102366_

Round 1

Reviewer 1 Report

Dear Authors and Editor,

The manuscript shows current and valuable subject about the Role of Responsible Packaging in Enhancing Naturalness and Healthiness Food Perceptions. However, more current references should be mentioned in this manuscript. For example Alhamdi (2020) studied role of packaging in consumer buying behavior. Below are just a few suggestions that were not quoted and would bring valuable content to the article. Please also specify what this research brings new and what is its purpose, because it is missing. The research methodology is rather well described and explained, but please describe in more detail the products and their packaging that were tested. Discussion section should be improved and additional references should be mentioned were possible.

Below some proposed references

Alhamdi, F. M. (2020). Role of packaging in consumer buying behavior. Management Science Letters10(6), 1191–1196. https://doi.org/10.5267/j.msl.2019.11.040

Suman Prosad Saha. (2020). Impact of Product Packaging on Consumer Buying Decision. Journal of Engineering and Science Research4(2), 17–22. https://doi.org/10.26666/rmp.jesr.2020.2.4

Wandosell, G., Parra-Meroño, M. C., Alcayde, A., & Baños, R. (2021, February 1). Green packaging from consumer and business perspectives. Sustainability (Switzerland). MDPI AG. https://doi.org/10.3390/su13031356

Otto, S., Strenger, M., Maier-Nöth, A., & Schmid, M. (2021, May 20). Food packaging and sustainability – Consumer perception vs. correlated scientific facts: A review. Journal of Cleaner Production. Elsevier Ltd. https://doi.org/10.1016/j.jclepro.2021.126733

Author Response

See attached file containing responses to all reviewers.

Reviewer 2 Report

The research is focused on the perception of food by consumers, relating mainly to the attributes of "natural" and "healthy'', and their influence on the purchase decision.
This work is interesting, from the perspective of product design efforts (across the breadth of the concept) and how it affects the consumer's purchase decision.
The methodology used, based on filtering concepts and associating perceptions relative to attributes such as natural or healthy, is basic but original.
The article is well written, the methodology is clear, and the results are interesting.
The authors indicate the limitations of the study, but the conclusions could be expanded, because although they respond to the hypothesis of the principle, they can be better developed.

I would like to ask the authors to expand the conclusions and define possible improvements to the study, as well as future lines of work.

Author Response

(The authors gave the same response as above.)
